# Survival study of cytoreductive prostatectomy for prostate cancer with lung metastasis: A propensity score matching study based on the SEER database

**An-Ping Xiang\*, Xue-Feng Yuan, Zhen-Qian Qin**

Department of Urology, Yixing People's Hospital, Yixing, People's Republic of China

\* anping108@163.com

## Abstract

### Objective

To investigate the specific tumor survival situation of cytoreductive prostatectomy for prostate cancer with lung metastasis.

### Methods

Case data of prostate cancer patients diagnosed from 2010 to 2015 were collected from the SEER database (https://seer.cancer.gov/) to obtain age, race, marital status, T stage, N stage, ISUP grade, prostate-specific antigen level (PSA), surgery, radiotherapy, chemotherapy, and survival time. The subjects were divided into surgical groups (CDS): resection of the primary lesion (prostate) and non-surgical groups (NCDS). The purpose of palliative debulking is to reduce the tumor burden. Only the primary lesion of the prostate is removed, while metastatic lesions are not subjected to surgical resection. Propensity score matching (PSM) was used to control confounding factors, and the survival outcomes of the two groups were systematically evaluated. Kaplan-Meier survival curves and forest plots were plotted to further verify the survival benefits of the surgery.

### Results

A total of 808,782 cases were screened through the SEER database, and ultimately 1548 cases were eligible for prostate cancer with lung metastasis, including 136 cases in the surgical group and 1412 cases in the non-surgical group. After PSM, there were 70 cases in the surgical group and 70 cases in the non-surgical group. The Kaplan-Meier survival curve showed no survival difference between the two groups, P = 0.28, and the forest plot further confirmed no survival difference between the two groups. The Kaplan-Meier survival curve analysis for the subgroups that received further radiotherapy and chemotherapy showed that there was no

**Data availability statement:** The data underlying the results presented in the study are available from SEER database. The dataset used in this analysis is titled "SEER Research Data, 2010–2015", and was accessed via SEER*Stat software version 8.4.5. The data can be searched and downloaded directly from the following URL:https://seer.cancer.gov/data-software/.

**Funding:** The author(s) received no specific funding for this work.

**Competing interests:** An-Ping Xiang designed the study and drafted and revised the manuscript, Xue-Feng Yuan and Zhen-Qian Qin analyzes the data and draw graphs.

statistically significant difference in the tumor-specific survival time among the sub-groups (P > 0.05).

## Conclusion

Cytoreductive prostatectomy does not prolong the specific tumor survival time of prostate cancer patients with lung metastasis, Radiotherapy and chemotherapy also fail to extend the specific tumor survival time for these patients.

---

## Introduction

Prostate cancer is one of the most common malignant tumors among men worldwide. According to the 2020 Global Cancer statistics, it ranks second among male malignant tumors in terms of incidence, with more than 1.4 million new cases and about 375,000 deaths each year [1]. The incidence of prostate cancer is on the rise as the population ages and prostate-specific antigen (PSA) screening becomes more widespread. Although early-stage prostate cancer patients have a good prognosis through radical surgery or radiotherapy, about 10–15 percent of patients have distant metastasis at first diagnosis [2], and another 20–30 percent of patients with localized prostate cancer eventually progress to metastatic prostate cancer after treatment [3]. Metastatic prostate cancer has a 5-year survival rate of less than 30 per cent and is the leading cause of prostate cancer-related deaths [4].

Among metastatic prostate cancers, bone metastases are the most common (about 80%), followed by lymph node metastases (40%) and visceral metastases (10%−15%) [5]. Lung metastases, as one of the main forms of visceral metastases, account for about 10–15 percent of metastatic cases [6,7]. Prostate cancer with lung metastasis tends to have more aggressive biological behavior compared to bone metastasis, as shown by higher Gleason scores, shorter PSA doubling times and greater resistance to conventional endocrine therapy [8]. In recent years, with the application of new imaging techniques such as PSMA PET-CT, the detection rate of lung metastases has significantly increased, presenting new challenges for the clinical management of prostate cancer with lung metastases.

The potential value of primary site surgery is particularly worth exploring for prostate cancer patients with lung metastases. Anatomically, lung metastases typically form through hematogenous dissemination, which may reflect more aggressive tumor biological characteristics [9]. From a clinical perspective, patients with lung metastases are often excluded from key clinical trials, resulting in a lack of evidence-based medical evidence. Current studies on surgical treatment for prostate cancer with lung metastases are mostly small-sample retrospective studies with inconsistent results. In a meta-analysis Mahmoud AM et al. revealed that prostate cancer with visceral metastasis usually predicts poor disease outcomes, yet the clinical progression of patients with visceral metastasis may vary greatly. Specifically, patients with lung metastases tend to have better disease progression compared to those with other visceral metastases. The prognosis and survival outcomes of prostate cancer lung

metastases vary significantly according to existing studies, with survival periods ranging from 3 months to 88 months, with a median survival of 26 months. The authors also noted in the study that surgical resection of prostate cancer lung metastases remains controversial [10].

The differences between these studies may stem from variations in patient selection, surgical indications, and confounding factor control. It is notable that most of the previous studies failed to adequately control selection bias in terms of research methodology. Patients who underwent surgery were typically younger, in better general condition, and had a lower tumor burden, and these factors themselves were associated with a good prognosis. Propensity Score Matching (PSM) is an effective statistical method that reduces selection bias in observational studies by constructing virtual randomized controls [11]. PSM has been used to evaluate the efficacy of local treatment for metastatic diseases in other tumors such as gastric cancer and renal cancer [12,13], but there is still a lack of research in prostate cancer, especially in patients with lung metastases.

Based on this background, this study aims to systematically evaluate the impact of primary site surgery on the survival prognosis of prostate cancer patients with lung metastasis by using large sample data from the US SEER database (2010–2015) and controlling confounding factors through a strict PSM method. To guide clinicians in developing optimal treatment strategies for specific patients, and to provide evidence-based support for the update of relevant guidelines.

## Materials and methods

### Data sources

This study employed a clinical retrospective analysis method to obtain data on prostate cancer patients last updated on March 5, 2025 at 18 registry sites in the SEER database through SEER*State8.4.5. Since the SEER database is a public database, there is no need for patients to sign informed consent forms or for institutional ethical reviews. A total of 808,782 cases of prostate cancer aged 60–79 were exported through the software. The data were cleaned and screened according to inclusion and exclusion criteria, resulting in 1548 cases of prostate cancer with lung metastasis, including 136 cases that underwent cytoreduced-tumor prostatectomy and 1412 cases that did not undergo surgery. After PSM, there were 70 cases in the surgical group and 70 cases in the non-surgical group. To ensure the correctness of the temporal sequence, in this analysis, all survival times are calculated from the date of diagnosis of lung metastasis, thereby avoiding the bias caused by including the time before metastasis in the survival analysis. Ethical review and approval were waived for this study, due to the data being publicly available and anonymous.

### The inclusion criteria included

Pathologically confirmed prostate adenocarcinoma (ICD-O-3 code 8140);

Imaging confirmed lung metastases, with or without bone metastases;

Complete recording of surgical status and survival data;

The age range is 60–79 years old.(≥ 60 years old is the high-risk group for prostate cancer. The group over 79 years old has relatively poorer physical condition. Therefore, for the age group of 60–79 years old, surgical treatment is necessary, and they can better withstand the trauma caused by the surgery. Therefore, the author chose the age group of 60–79 years old as the research subjects).

### Exclusion criteria

Liver metastases or incomplete metastasis information;

With brain metastases or incomplete transfer information;

Combined with other malignant tumors, prostate cancer is not the only primary tumor;

Non-tumor-specific death patients.

### Study variables

The age of prostate cancer patients collected from the SEER database (60–64 years, 65–69 years, 70–74 years, 75–79 years), race (Black, White, Other, Unknown), marital status (Divorced, Married (including common law), Single (never married, Widowed, Other/Unknown), T stage (T0, T1, T2, T3, T4, Tx/NA), N stage (N0, N1, Nx/NA), ISUP grade score (2, 3, 4, 5, Unknown), prostate-specific antigen level (PSA), surgery, Clinical information such as radiotherapy, chemotherapy, survival time, etc.

### Research data processing and statistical methods

Data of PCa patients diagnosed between 2010 and 2015 were collected from the SEER database. All eligible data were analyzed using SPSS (V21.0) and R software (V4.4.2). Count data were expressed as the number of cases (percentage), and chi-square test was used for comparison between groups to describe the differences between the two groups of variables. $P < 0.05$ indicates a statistically significant difference.

### Propensity score matching (PSM)

Propensity score matching (PSM) was used in this study to control selection bias and to make the baseline of the surgical group (CDS) comparable to that of the non-surgical group (NCDS). A binary logistic regression model was used to calculate the propensity score for patients to undergo surgery (CDS). Baseline covariables included in the model were: age (group), race, marital status, T stage, N stage, ISUP grade, PSA at diagnosis, whether radiotherapy was received, and whether chemotherapy was received. 1:1 nearest neighbor matches were made between the surgical group and the non-surgical group. Matching was performed on a logarithmic scale of propensity scores, and caliper values were set at 0.7 times standard deviation to improve matching quality and comparability. After matching, standardized mean differences (SMD) were primarily used to assess the balance of all covariates between groups, supplemented by chi-square tests and t-tests for statistical differences. After matching, the standardized mean differences (SMD) of all covariates were all less than 0.1, indicating good balance between the groups. The matching effect was presented through baseline characteristic comparison tables (as shown in Table 1) and propensity score distribution maps (as shown in Fig 1). All survival analyses (including Kaplan-Meier curves, Log-rank tests, median survival time comparisons) were based on PSM-matched cohort to evaluate the net effect of surgery on survival.

## Results

### The selection and propensity scores of the study cohort matched

The SEER database (2010–2015) screened 808,782 prostate cancer patients and ultimately included 1548例 (0.19%) patients with lung metastases, including 136例 (8.78%) in the surgical group (CDS) and 1412例 (91.22%) in the non-surgical group (NCDS).

Before matching, there were significant differences between the CDS group and the NCDS group in T stage (P = 0.0357), ISUP grade (P < 0.001), and radiotherapy (P = 0.013) (Table 1). Using a 1:1 nearest neighbor propensity score match (caliper value = 0.7), 70 pairs of patients were successfully matched (CDS: 70cases, NCDS:70 cases). After matching, all baseline covariates (age, race, marital status, T/N/ISUP stage, PSA, chemoradiotherapy) were well balanced (standardized mean difference SMD < 0.1). The intergroup differences in T stage, ISUP grade, radiotherapy, which had previously been different, disappeared (P > 0.05). Baseline characteristics before and after PSM are shown in Table 1.

**Table 1. Baseline characteristics before and after PSM, showing statistical comparisons of t-tests or χ² tests between CDS groups and NCDS groups.**

| parameter | Pre-PSM | | | | Post-PSM | | | |
|---|---|---|---|---|---|---|---|---|
| | CDS (136) | NCDS (1412) | χ²/T value | P value | CDS (70) | NCDS (70) | χ²/T value | P value |
| **Age** | | | 4.1967 | 0.2209 | | | 3.0329 | 0.4087 |
| 60–64 years | 33 (24.3%) | 329 (23.3%) | | | 20 (28.6%) | 17 (24.3%) | | |
| 65–69 years | 27 (19.9%) | 391 (27.7%) | | | 14 (20.0%) | 13 (18.6%) | | |
| 70–74 years | 38 (27.9%) | 361 (25.6%) | | | 21 (30.0%) | 16 (22.9%) | | |
| 75–79 years | 38 (27.9%) | 331 (23.4%) | | | 15 (21.4%) | 24 (34.3%) | | |
| **Race** | | | 5.139 | 0.1065 | | | 0.30849 | 0.9558 |
| Black | 17 (12.5%) | 281 (19.9%) | | | 6 (8.6%) | 7 (10.0%) | | |
| White | 101 (74.3%) | 990 (70.1%) | | | 53 (75.7%) | 54 (77.1%) | | |
| Other | 16 (11.8%) | 127 (9.0%) | | | 10 (14.3%) | 8 (11.4%) | | |
| Unknown | 2 (1.5%) | 14 (1.0%) | | | 1 (1.4%) | 1 (1.4%) | | |
| **Marital** | | | 3.0808 | 0.5901 | | | 4.0341 | 0.4146 |
| Divorced | 17 (12.5%) | 147 (10.4%) | | | 10 (14.3%) | 7 (10.0%) | | |
| Married (including common law) | 85 (62.5%) | 813 (57.6%) | | | 45 (64.3%) | 48 (68.6%) | | |
| Single (never married) | 21 (15.4%) | 267 (18.9%) | | | 9 (12.9%) | 4 (5.7%) | | |
| Widowed | 6 (4.4%) | 80 (5.7%) | | | 2 (2.9%) | 4 (5.7%) | | |
| Other/Unknown | 7 (5.1%) | 105 (7.4%) | | | 4 (5.7%) | 7 (10.0%) | | |
| **T** | | | 12.836 | 0.0357 | | | 3.4593 | 0.7132 |
| T0 | 0 (0%) | 6 (0.4%) | | | 0 (0%) | 2 (2.9%) | | |
| T1 | 13 (9.6%) | 67 (4.7%) | | | 5 (7.1%) | 4 (5.7%) | | |
| T2 | 6 (4.4%) | 118 (8.4%) | | | 5 (7.1%) | 3 (4.3%) | | |
| T3 | 7 (5.1%) | 32 (2.3%) | | | 5 (7.1%) | 3 (4.3%) | | |
| T4 | 5 (3.7%) | 59 (4.2%) | | | 3 (4.3%) | 2 (2.9%) | | |
| Tx/NA | 105 (77.2%) | 1130 (80.0%) | | | 52 (74.3%) | 56 (80.0%) | | |
| **N** | | | 0.67466 | 0.7198 | | | 0.32275 | 0.8777 |
| N0 | 15 (11.0%) | 182 (12.9%) | | | 9 (12.9%) | 9 (12.9%) | | |
| N1 | 15 (11.0%) | 133 (9.4%) | | | 8 (11.4%) | 6 (8.6%) | | |
| Nx/NA | 106 (77.9%) | 1097 (77.7%) | | | 53 (75.7%) | 55 (78.6%) | | |
| **ISUP** | | | 67.472 | < 0.001 | | | 11.47 | 0.0632 |
| 2 | 4 (2.9%) | 9 (0.6%) | | | 3 (4.3%) | 3 (4.3%) | | |
| 3 | 3 (2.2%) | 22 (1.6%) | | | 0 (0%) | 8 (11.4%) | | |
| 4 | 15 (11.0%) | 139 (9.8%) | | | 9 (12.9%) | 8 (11.4%) | | |
| 5 | 54 (39.7%) | 211 (14.9%) | | | 27 (38.6%) | 17 (24.3%) | | |
| Unknown | 60 (44.1%) | 1031 (73.0%) | | | 31 (44.3%) | 34 (48.6%) | | |
| **PSA** | | | 379.66 | 0.6501 | | | 122 | 0.3573 |
| Mean (SD) | 24.5 (24.4) | 34.3 (26.2) | | | 24.7 (24.5) | 29.2 (25.5) | | |
| Median [Min, Max] | 15.1 [0.300, 91.9] | 26.8 [0.500, 97.2] | | | 15.2 [0.300, 91.9] | 19.7 [0.500, 96.6] | | |
| Missing | 65 (47.8%) | 912 (64.6%) | | | | | | |
| **Radiation** | | | 5.5941 | 0.013 | | | 0.525 | 0.4695 |
| Yes | 19 (14.0%) | 328 (23.2%) | | | 8 (11.4%) | 12 (17.1%) | | |
| No/Unknown | 117 (86.0%) | 1084 (76.8%) | | | 62 (88.6%) | 58 (82.9%) | | |
| **Chemotherapy** | | | 0.64556 | 0.3954 | | | 0.7524 | 0.8407 |
| Yes | 36 (26.5%) | 325 (23.0%) | | | 17 (24.3%) | 15 (21.4%) | | |
| No/Unknown | 100 (73.5%) | 1087 (77.0%) | | | 53 (75.7%) | 55 (78.6%) | | |

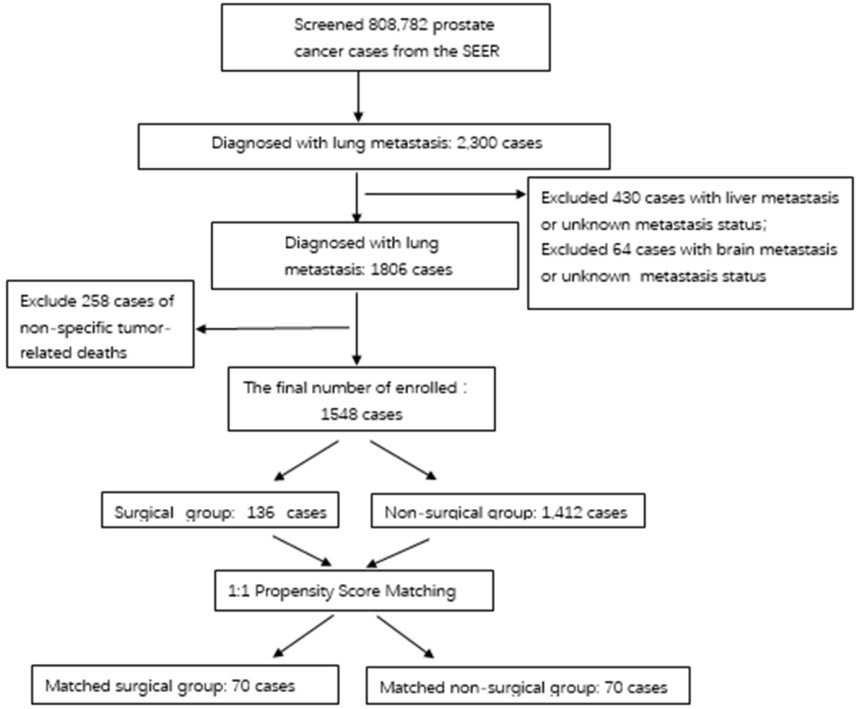

**Fig 1. Distribution map of propensity scores between the CDS group and the NCDS group before and after propensity score matching.**

## Survival analysis and median survival time

Based on the comparable cohort after PSM matching (CDS group: n = 70, NCDS group: n = 70), survival analysis was performed: Kaplan-Meier survival curve analysis showed (Fig 2) that the survival curves of the surgical group (CDS) and the non-surgical group (NCDS) highly overlapped, indicating that the survival trends of the two groups of patients were highly consistent. Log-rank test result: P = 0.28. The result clearly indicated that there was no statistically significant difference in Overall Survival (OS) between the two groups of patients. The median survival time in the CDS group (surgical group) was 44 months (95% confidence interval [CI]: 30–65 months). The median survival time in the NCDS group (non-surgical group) was 58 months (95% CI: 32–85 months), and there was no statistically significant difference in median survival time between the two groups (P > 0.05). There was a broad overlap of 95% confidence intervals between the two groups (CDS: 30–65 months vs. NCDS: 32–85 months), further supporting the conclusion that there was no significant difference (Fig 3). Median survival time after PSM is shown in Fig 3.

## Subgroup analysis based on PSM data

In the analysis of the radiotherapy subgroup, the patients were divided into four subgroups based on their surgical status and radiotherapy reception. The Kaplan-Meier survival analysis showed that there was no statistically significant difference in tumor-specific survival among the groups (p = 0.19) (see Fig 4); in the chemotherapy subgroup analysis, the patients were divided into four subgroups based on their surgical status and chemotherapy reception. The Kaplan-Meier analysis showed that there was no statistically significant difference in tumor-specific survival among the groups (p = 0.083) (see Fig 5).

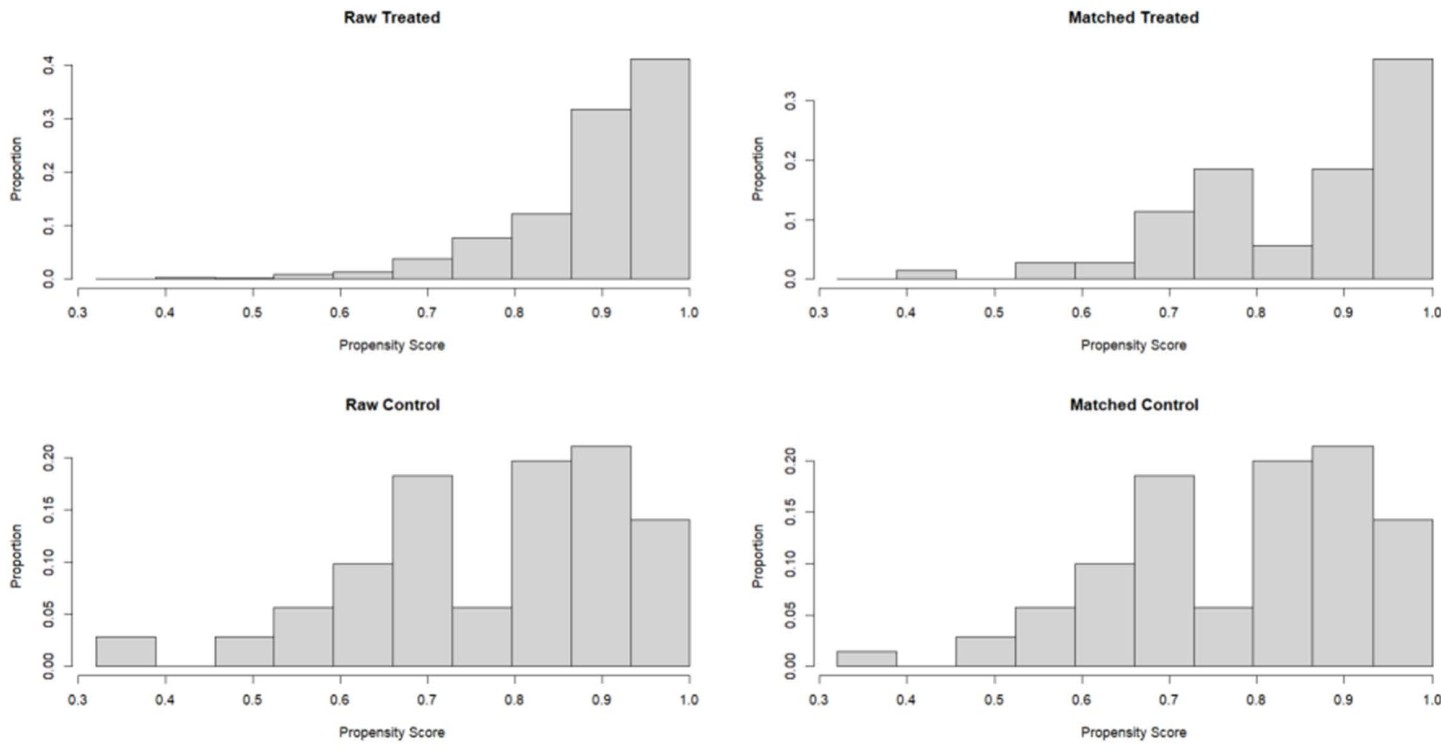

**Fig 2. Survival curves of the two groups after PSM.**

## Discussion

### 1. Comparative analysis of research results with existing literature

In recent years, with the intensification of global population aging and the widespread use of prostate-specific antigen (PSA) screening, the incidence of prostate cancer has continued to rise [14]. According to the 2020 Global cancer statistics, there were more than 1.4 million new cases of prostate cancer, about 375,000 deaths, and the absolute number of metastatic cases increased [1,15]. Prostate cancer metastases have a significant organ tendency, with bone metastases being the most common, reaching about 80 percent of metastases, often found in axial bones such as the spine and pelvis, and often spread through local infiltration or the lymphatic system. Next is lymph node metastasis, with an incidence of about 40 percent, mainly involving pelvic and retroperitoneal lymph nodes [16,17]. Visceral metastases account for 10–15 percent, with lung metastases being the most common form of visceral metastases, accounting for about 10–15 percent of metastatic prostate cancer [18]. It is worth noting that there are essential differences in the biological behavior between lung metastases and bone metastases: The former is dominated by hematogenous dissemination, often accompanied by higher tumor grades (35.4% in the surgical group in this study), shorter PSA doubling times, and stronger resistance to endocrine therapy, suggesting more aggressive biological characteristics [10,19].

Based on the large sample data from the SEER database, the analysis showed that the incidence of prostate cancer with lung metastasis in patients aged 60–79 was approximately 0.23% (1806/808782), of which only 8.9% (161/1806) underwent primary site resection. Controlling for confounding factors by propensity score matching (PSM) method, the impact of cytoreductive prostatectomy on survival in patients with prostate cancer with lung metastasis was systematically evaluated. The results showed no statistically significant difference in survival time between the surgical group (CDS) and the non-surgical group (NCDS), either before or after PSM (before PSM: P > 0.05; After PSM: P = 0.79), median survival

**Fig 3. Median survival time of the two groups after PSM.**

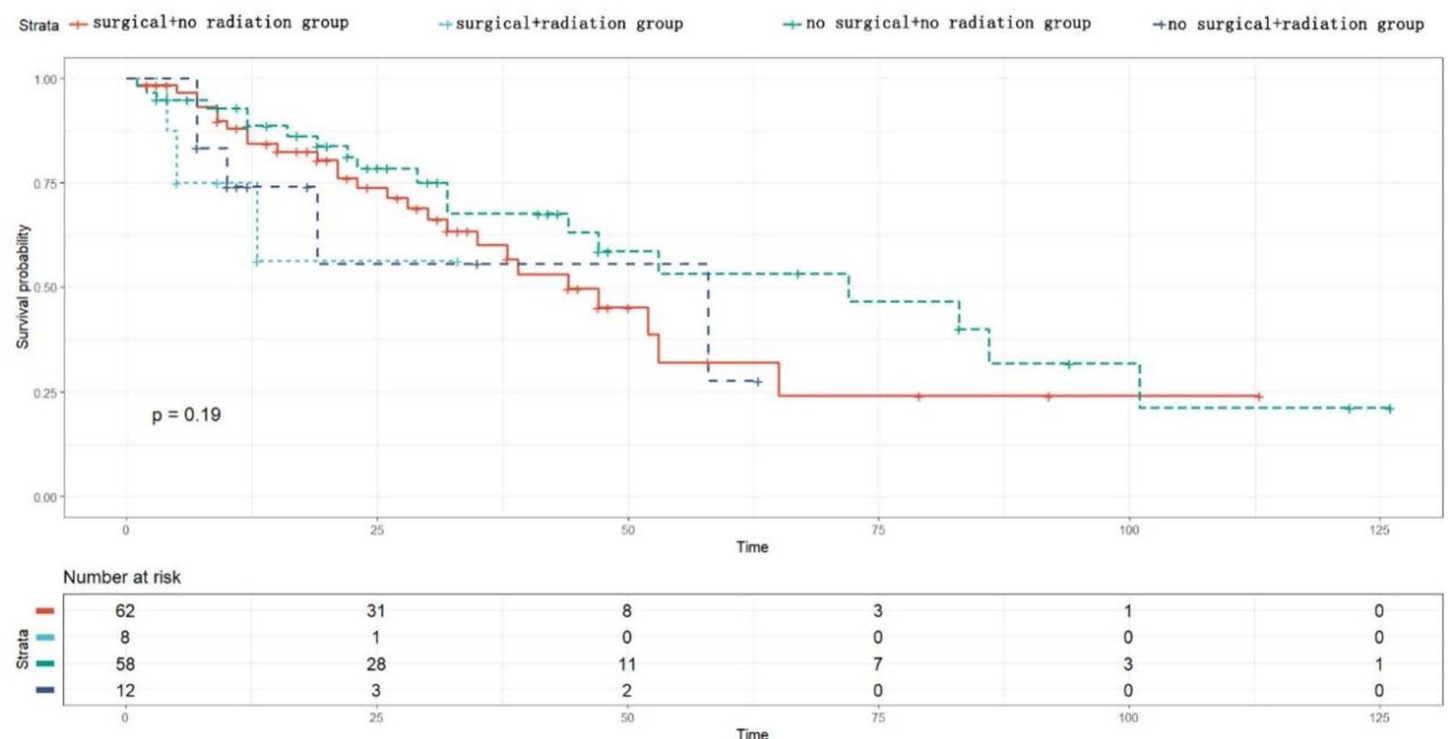

**Fig 4. Subgroup survival analysis chart for radiotherapy.**

time was 39 months and 35 months, respectively, with no significant difference between the two groups. This study balanced key prognostic factors such as age, race, tumor stage, and PSA level through the PSM method, reduced selection bias, and the results were highly credible.

The risk of tumor metastasis is closely associated with biological characteristics: Patients with high ISUP grades (particularly grade 5), high PSA levels (median PSA was 15.3 ng/mL in the surgical group and 26.3 ng/mL in the non-surgical group in this study), and short PSA doubling times were more likely to develop metastasis. The risk of visceral metastases, such as lung metastases, was significantly increased [17,20,21]. In addition, therapeutic factors also influence the progression of metastasis: early localized prostate cancer without standardized treatment (such as radical surgery or radiotherapy) increases the risk of progression to metastatic disease by 20–30 percent [22,23].

## 2. Implications of epidemiological characteristics for clinical decision-making

This study confirmed through a large sample analysis that cytoreduction-induced prostatectomy failed to improve survival outcomes in patients with lung metastasis (no statistically significant difference in survival between the two groups after PSM, P = 0.79), Subgroup survival analyses for the two groups after further radiotherapy and chemotherapy showed that there was no statistically significant difference in tumor-specific survival among the subgroups (P > 0.05). which is directly related to the epidemiological pattern of prostate cancer metastasis: Lung metastasis, as a result of hematogenous dissemination, often indicates that the systemic tumor burden is already at a high level, and a single local treatment is difficult to reverse the disease progression. In contrast, patients with bone metastases may benefit from resection of the primary lesion as the main route of metastasis is local spread. However, systemic treatment (such as new endocrine therapy) should still be the first choice for patients with lung metastases [9,24].

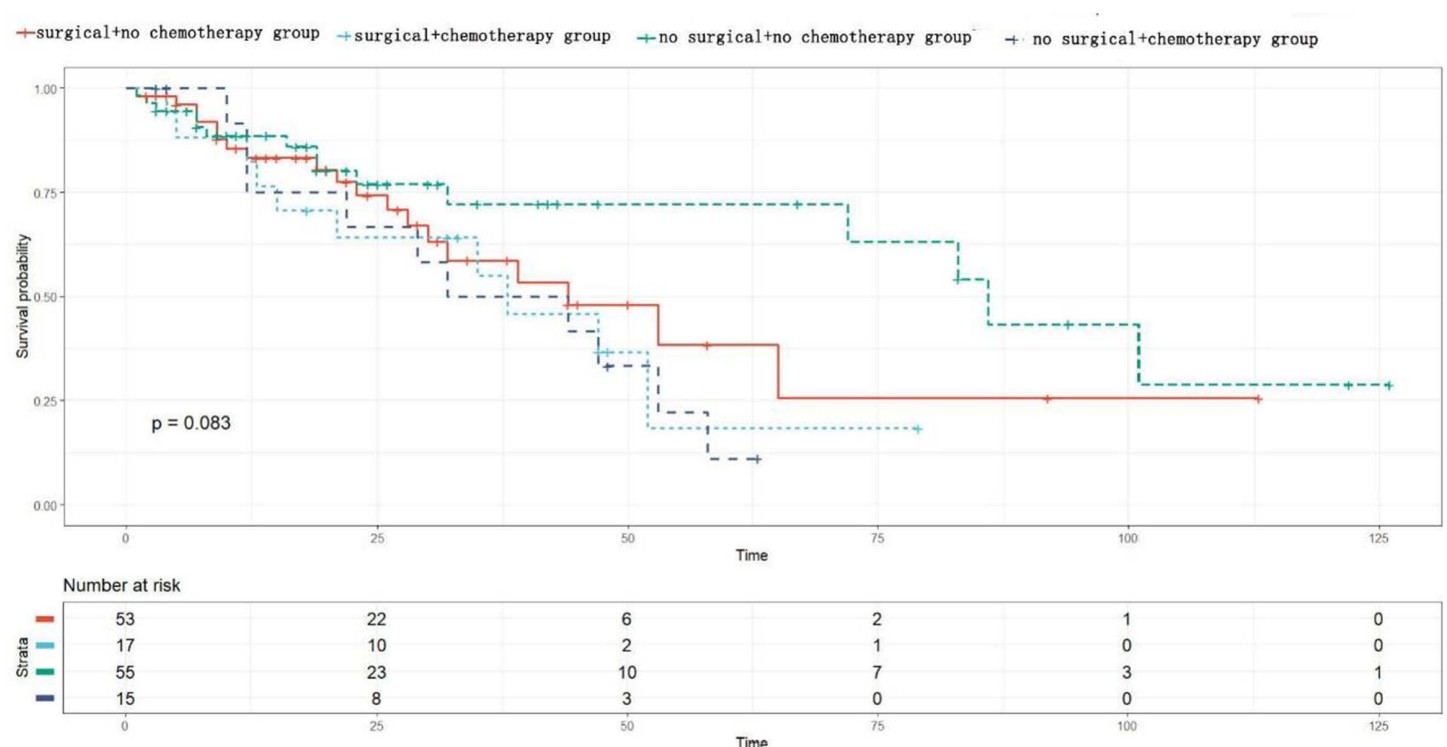

**Fig 5. Subgroup survival analysis chart for chemotherapy.**

The limitations of current epidemiological studies also provide clues for future directions: The SEER database lacks data on tumor molecular characteristics (such as AR-V7, DNA repair gene defects) and the use of new therapeutic drugs, making it difficult to precisely identify high-risk metastasis-risk subgroups. In the future, molecular epidemiological studies need to be combined to explore the differences between lung and bone metastases in terms of gene mutation profiles and activation of signaling pathways, providing targets for metastasis prevention and individualized treatment [25,26].

However, for patients with oligosases (such as single lung metastases) or those with a heavy primary lesion burden accompanied by local symptoms (such as hematuria, dysuria), resection of the primary lesion may have palliative value [27–29]. In this study, there was a difference in radiotherapy use between the surgical group and the non-surgical group (P = 0.01495 before PSM), suggesting that clinicians can selectively combine local radiotherapy for surgical patients, but the effectiveness of this combination strategy still needs to be verified by high-quality studies.

### 3. Advantages and limitations of the study methodology

The main advantage of this study lies in the use of a large national sample of data from the SEER database, covering prostate cancer patients at 18 registration sites in the United States, which is highly representative. Through a rigorous PSM approach, baseline characteristics of the surgical group and the non-surgical group were successfully balanced, especially key confounding factors such as T stage, ISUP grade, and radiotherapy, making the two groups comparable and enabling a more accurate assessment of the real efficacy of the surgery. In addition, detailed clinicopathological characteristics, including age, race, marital status, PSA level, etc. were included in the study, and a comprehensive analysis of factors that might affect survival was conducted. There are only a few previous literature reports on the treatment of

isolated lung metastases after radical prostatectomy [30,31], and there are currently no research reports on surgical treatment of prostate cancer with lung metastases. Our study fills this gap.

However, there are certain limitations in this study. First, the SEER database lacks key information such as surgical methods (open surgery vs.) Factors such as postoperative complications, specific chemotherapy regimens, and the use of new endocrine therapy drugs, which may affect survival outcomes, cannot be adjusted in the analysis. Secondly, the assessment of lung metastases in the database relies on imaging examinations, which may lead to missed diagnoses or misdiagnoses, especially the limited ability to detect small metastases. In addition, the study subjects were limited to patients aged 60–79, excluding the younger or older population, so caution is needed when extrapolating the results to other age groups. Finally, as a retrospective observational study, despite the use of the PSM approach, it was not possible to completely eliminate the influence of residual confounding factors such as the patient's physical condition (ECOG score), tumor molecular characteristics (such as AR-V7 expression, DNA repair gene defect), complications, PSA doubling time and specific systemic therapy which may affect surgical decisions and prognosis, but were not recorded in the database. The sample size of this study is not large, so the accuracy of the conclusion requires support from more research.

## Conclusions

The large-sample analysis based on the SEER database in this study shows that prostate cancer metastasis, especially lung metastasis, has unique biological characteristics and treatment challenges. This study did not show association between cancer specific survival benefit and cytoreductive prostatectomy in 60–79 years old patients with lung metastases. The same can be stated for chemotherapy and radiotherapy. In the future, precision medicine will be needed to deeply analyze the mechanisms of metastasis, identify potential beneficiaries, and develop new therapies to improve patient prognosis. In clinical practice, individualized comprehensive treatment plans should be developed based on the patient's tumor burden, molecular characteristics and overall condition, avoiding excessive reliance on local treatment.

## Author contributions

**Conceptualization:** An-Ping Xiang.

**Data curation:** An-Ping Xiang.

**Formal analysis:** Zhen-Qian Qin.

**Methodology:** Xue-Feng Yuan.

**Software:** An-Ping Xiang.

**Writing – original draft:** Xue-Feng Yuan, Zhen-Qian Qin.

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
