## [Decision Letter · Decision Letter 0]

12 Oct 2025

Dear Dr. anping,

Thank you for submitting your manuscript to PLOS ONE. After careful consideration, we feel that it has merit but does not fully meet PLOS ONE’s publication criteria as it currently stands. Therefore, we invite you to submit a revised version of the manuscript that addresses the points raised during the review process.

We look forward to receiving your revised manuscript.

Kind regards,

Li Yang, M.D.

Academic Editor

PLOS ONE

Journal Requirements:

“An-Ping Xiang designed the study and drafted and revised the manuscript, Xue-Feng Yuan and Zhen-Qian Qin analyzes the data and draw graphs.”

3. Please note that your Data Availability Statement is currently missing a direct link to access each database. If your manuscript is accepted for publication, you will be asked to provide these details on a very short timeline. We therefore suggest that you provide this information now, though we will not hold up the peer review process if you are unable.

Additional Editor Comments:

Thanks for submitting your work to PLOS ONE. Your manuscript has now been assessed by our editorial team and external peer experts. While they found it interesting, you will see that they have raised many serious problems and are advising that you revise your manuscript thoroughly. At the same time, please submit the point-by-point responses to reviewers' comments. If you are prepared to undertake the work required, I would be pleased to reconsider my decision. Please note that this revision decision does not assure the acceptance of your work. Thanks for the opportunity to consider your work.

Reviewers' comments:

Reviewer's Responses to Questions

**Comments to the Author**

1. Is the manuscript technically sound, and do the data support the conclusions?

Reviewer #1: Partly

Reviewer #2: No

Reviewer #3: No

Reviewer #4: Yes

2. Has the statistical analysis been performed appropriately and rigorously?

Reviewer #1: I Don't Know

Reviewer #2: No

Reviewer #3: No

Reviewer #4: Yes

3. Have the authors made all data underlying the findings in their manuscript fully available?

Reviewer #1: Yes

Reviewer #2: Yes

Reviewer #3: No

Reviewer #4: Yes

4. Is the manuscript presented in an intelligible fashion and written in standard English?

Reviewer #1: Yes

Reviewer #2: Yes

Reviewer #3: No

Reviewer #4: Yes

Reviewer #1: This study represents a valuable contribution to the field of urologic oncology, addressing a clinically relevant question regarding the role of cytoreductive prostatectomy in prostate cancer patients with lung metastasis. The use of a large, population-based dataset from the SEER database and the application of propensity score matching to minimize selection bias are significant strengths. The manuscript is well-structured, the methodology is sound, and the conclusions are clearly supported by the data. The topic is timely and of great interest to clinicians seeking to optimize treatment strategies for metastatic prostate cancer. However, to further strengthen the manuscript, the following revisions are suggested:

1.

The final matched cohort consists of only 79 patients per group. While PSM helps balance covariates, such a small sample may limit the statistical power to detect clinically meaningful differences. Please discuss this limitation and consider performing a power calculation or sensitivity analysis.

2.

The term “reductive prostatectomy” is ambiguous. Please clarify whether this refers to radical prostatectomy, cytoreductive surgery, or palliative debulking. Specify the surgical intent and extent in the Methods section.

3.

Key prognostic factors such as ECOG performance status, comorbidities, PSA doubling time, and specific systemic therapies (e.g., AR-targeted therapies, chemotherapies) are not accounted for. These omissions may introduce residual confounding. Please acknowledge this limitation.

4.

The study does not distinguish between oligometastatic and polymetastatic disease. Patients with solitary lung metastasis may behave differently than those with multiple metastases. Subgroup analysis based on metastatic burden would strengthen the conclusions.

5.

The analysis is based on overall survival. Including cancer-specific survival might provide more direct insight into the effect of surgery on disease progression.

6.

The abstract mentions “1,8006” cases, which appears to be a typo (likely 1,806). Please verify all numerical values throughout the manuscript for consistency.

7.

The study is restricted to patients aged 60–79 years. This limits the applicability of findings to younger or older populations. Please discuss this explicitly.

8.

The type, timing, and dosage of radiotherapy and chemotherapy are not detailed. These treatments likely influence survival and should be considered in the analysis or at least discussed as potential confounders.

9.

The caliper value of 0.02 is quite strict and may have excluded many patients. Justify this choice or consider using a more conventional caliper (e.g., 0.05–0.10) to improve matching efficiency.

10.

The conclusion states that cytoreductive prostatectomy “does not prolong survival,” but the study only shows no statistical difference. Consider rephrasing to reflect that no survival benefit was observed in this cohort, rather than a definitive absence of benefit.

Reviewer #2: 1.Limited to a Specific Age Group: The study only included prostate cancer patients aged 60 to 79, which limits the external validity of the results (i.e., whether the results can be applied to other age groups). Younger or older patients may have different survival prognoses, so the study's conclusions cannot be directly applied to these groups.

2.Lack of Stratified Analysis: The absence of stratified analysis is an important limitation of the study. Stratified analysis could assess the treatment effects based on different patient characteristics (such as age, PSA levels, tumor staging, ISUP grading, etc.), helping to identify subgroups of patients who may benefit from specific treatments. Although the study used propensity score matching (PSM) to reduce selection bias, it did not conduct further stratified analysis, making it impossible to explore whether different patient subgroups have different survival benefits after undergoing prostatectomy.

3.Lack of Multivariable Regression Analysis: The absence of multivariable regression analysis is another limitation. Including multivariable regression analysis could help further quantify the independent effects of surgical treatment and other clinical factors on the survival of prostate cancer patients, providing more detailed statistical evidence. This would not only improve the precision of the study but also offer more meaningful guidance for clinical treatment.

Reviewer #3: This retrospective SEER analysis evaluates whether cytoreductive prostatectomy (cRP) confers a survival benefit in men aged 60–79 with prostate cancer and lung metastasis (2010–2015). After 1:1 PSM (79 vs 79), the Kaplan–Meier comparison shows no OS difference (log-rank P=0.79), and the manuscript concludes cRP does not prolong survival in this subgroup. The clinical question is relevant; however, reporting inconsistencies, methodological limitations (time-related bias, missingness handling, PSM details), and presentation issues currently preclude reliable inference.

Reviewer #4: Nice, well designed and soundly modeled retrospective study on SEER data. Since the study is not controlled trial, I would recommend to authors to downplay their expression in Conclusion section when they talk about how the study confirmed non suitability of surgery in patients with lung metastasis, unfortunately this kind of studies can only suggest relationships, not confirm them.

**Do you want your identity to be public for this peer review?** For information about this choice, including consent withdrawal, please see our Privacy Policy

Reviewer #1: **Yes:** Mohammad Ebrahimnezhad

Reviewer #2: No

Reviewer #3: **Yes:** Yuyi Ou

Reviewer #4: **Yes:** Benjamin Benzon

---

## [Author Response · Author response to Decision Letter 1]

4 Nov 2025

I have provided detailed responses to all the questions in the "respond to reviewers" document.

---

## [Decision Letter · Decision Letter 1]

16 Dec 2025

Dear Dr. anping,

Thank you for submitting your manuscript to PLOS ONE. After careful consideration, we feel that it has merit but does not fully meet PLOS ONE’s publication criteria as it currently stands. Therefore, we invite you to submit a revised version of the manuscript that addresses the points raised during the review process.

We look forward to receiving your revised manuscript.

Kind regards,

Li Yang, M.D.

Academic Editor

PLOS One

Journal Requirements:

Additional Editor Comments (if provided):

Please further address reviewer comments.

Reviewers' comments:

Reviewer's Responses to Questions

**Comments to the Author**

Reviewer #1: All comments have been addressed

Reviewer #3: (No Response)

Reviewer #4: All comments have been addressed

2. Is the manuscript technically sound, and do the data support the conclusions?

Reviewer #1: Yes

Reviewer #3: (No Response)

Reviewer #4: Yes

3. Has the statistical analysis been performed appropriately and rigorously?

Reviewer #1: Yes

Reviewer #3: (No Response)

Reviewer #4: Yes

4. Have the authors made all data underlying the findings in their manuscript fully available?

Reviewer #1: Yes

Reviewer #3: (No Response)

Reviewer #4: Yes

5. Is the manuscript presented in an intelligible fashion and written in standard English?

Reviewer #1: Yes

Reviewer #3: (No Response)

Reviewer #4: Yes

Reviewer #1: (No Response)

Reviewer #3: (No Response)

Reviewer #4: Please rephrase the following text: "This study demonstrates that in patients aged 60 to 79,cytoreductive prostatectomy failed to prolong cancer-specific survival in prostate cancer patients with lung metastases, nor did

the combination of chemotherapy and radiotherapy extend cancer-specific survival." in to something like this: "This study did not show association between cancer specific survival benefit and cytoreductive prostatectomy in 60 to 79 years old patients with lung metastases. The same can be stated for chemotherapy and radiotherapy."

**Do you want your identity to be public for this peer review?** For information about this choice, including consent withdrawal, please see our Privacy Policy

Reviewer #1: No

Reviewer #3: No

Reviewer #4: **Yes:** Benjamin Benzon

---

## [Author Response · Author response to Decision Letter 2]

18 Dec 2025

Our response The reviewers did not have any recommendation requirements in this regard for my article.

Our response Thank you very much, editor, for checking the article. I have checked all the reference file formats and corrected the errors.

Reviewer #4: Please rephrase the following text: "This study demonstrates that in patients aged 60 to 79,cytoreductive prostatectomy failed to prolong cancer-specific survival in prostate cancer patients with lung metastases, nor did the combination of chemotherapy and radiotherapy extend cancer-specific survival." in to something like this: "This study did not show association between cancer specific survival benefit and cytoreductive prostatectomy in 60 to 79 years old patients with lung metastases. The same can be stated for chemotherapy and radiotherapy."

Our response Thank you for the valuable comments of the reviewers. I have revised my conclusion in the article.

---

## [Decision Letter · Decision Letter 2]

28 Jan 2026

Survival study of  cytoreductive prostatectomy for prostate cancer with lung metastasis: A propensity score matching study based on the SEER database

PONE-D-25-46062R2

Dear Dr. anping,

We’re pleased to inform you that your manuscript has been judged scientifically suitable for publication and will be formally accepted for publication once it meets all outstanding technical requirements.

Kind regards,

Li Yang, M.D.

Academic Editor

PLOS One

Additional Editor Comments (optional):

Thanks for the authors' efforts to comprehensively improve your manuscript according to editor's and reviewers' comments. I am pleased to inform you that your paper can be accepted for publication now.

Reviewers' comments:

Reviewer's Responses to Questions

**Comments to the Author**

Reviewer #4: All comments have been addressed

2. Is the manuscript technically sound, and do the data support the conclusions?

Reviewer #4: Yes

3. Has the statistical analysis been performed appropriately and rigorously?

Reviewer #4: Yes

4. Have the authors made all data underlying the findings in their manuscript fully available?

Reviewer #4: Yes

5. Is the manuscript presented in an intelligible fashion and written in standard English?

Reviewer #4: Yes

Reviewer #4: All of my comments have been addressed and I recommend the manuscript for publication, congratulations to authors.

**Do you want your identity to be public for this peer review?** For information about this choice, including consent withdrawal, please see our Privacy Policy

Reviewer #4: **Yes:** Benjamin Benzon

---

## [Editor Report · Acceptance letter]

PONE-D-25-46062R2

PLOS One

Dear Dr. anping,

I'm pleased to inform you that your manuscript has been deemed suitable for publication in PLOS One. Congratulations! Your manuscript is now being handed over to our production team.

Kind regards,

on behalf of

Dr. Li Yang

Academic Editor

PLOS One